# Nutritional, Phytochemical Characteristics and In Vitro Effect on α-Amylase, α-Glucosidase, Lipase, and Cholinesterase Activities of 12 Coloured Carrot Varieties

**DOI:** 10.3390/foods10040808

**Published:** 2021-04-09

**Authors:** Emel Yusuf, Aneta Wojdyło, Jan Oszmiański, Paulina Nowicka

**Affiliations:** Department of Fruit, Vegetable and Nutraceutical Plant Technology, Wrocław University of Environmental and Life Sciences, 37 Chełmońkiego Street, 51-630 Wrocław, Poland; emel.hasan.yusuf@upwr.edu.pl (E.Y.); aneta.wojdylo@upwr.edu.pl (A.W.); jan.oszmianski@upwr.edu.pl (J.O.)

**Keywords:** coloured carrots, phenolic acids, procyanidins, anthocyanins, carotenoids, enzyme inhibition effect

## Abstract

Twelve carrot varieties with different colours (purple, orange, yellow, and white) and sizes (normal, mini, and micro) were analysed for prospective health benefits (activities against diabetes-, obesity-, and aging- related enzymes—α-amylase, α-glucosidase, lipase, acetylocholinesterase, and butyrylocholinesterase, respectively) and nutritional contents (polyphenols, carotenoids, and chlorophylls). The conducted studies showed that the highest content of total polyphenols was observed in different sizes of purple carrots. The normal yellow and mini orange carrots demonstrated the highest content of carotenoids. According to the study results, the mini purple carrot showed the highest activities against diabetes-related enzyme (α-glucosidase); furthermore, the highest activities of cholinesterase inhibitors were observed for micro purple carrot. Nevertheless, normal orange carrot exhibited the highest activity against lipase. The results of the present study showed that purple-coloured carrot samples of different sizes (normal, mini, and micro) exhibited attractive nutritional contents. However, their pro-health effects (anti-diabetic, anti-obesity, anti-aging) should not be seen in the inhibition of amylase, glucosidase, lipase, and cholinesterase. Probably the mechanisms of their action are more complex, and the possible health-promoting effect results from the synergy of many compounds, including fibre, phytochemicals, vitamins, and minerals. Therefore, it would be worth continuing research on different varieties of carrots.

## 1. Introduction

*Daucus carota* L. is an *Apiaceae* member and grows in Europe, Asia, Africa, and Macaronesia. The cultivated carrot has evolved from the crossing of *D. carota* ssp. *carota* and *D. carota* ssp. *maximus* [1]. Interestingly, the first cultivated carrots were purple and yellow, followed by white carrots, and currently, orange carrots are the most popular in the world. Carrot is one of the top 10 consumable vegetables around the world, with high market share and nutrient values. One hundred grams of carrot provides approximately 41 kcal energy, 0.93 g protein, 9.58 g carbohydrates, and 2.8 g fibre [2,3].

Cultivated carrots are grouped by root colour, sugar-carotenoid content, and root shape, which are affected by development period, temperature during this time, and fertilizers [4,5]. Another reason for colour change in carrots is carotenoids, which in young carrots begin to accumulate after their first month of growth and is maintained about until the secondary growth is concluded [6].

Carrot varieties range from 5 cm to 50 cm for their root lengths, which are an important parameter for the marketing of carrot cultivars. In addition, cultivated carrots are classified as eastern and western carrots. Eastern carrots are purple and yellow coloured with branched roots; western carrots are orange, red, and white with unbranched roots. In this case, eastern carrot cultivars are rich in anthocyanins and western carrots are abundant in carotenes [7]. Nevertheless, purple carrot is twice as rich in α- and β-carotene contents as orange carrot; moreover, purple carrot possesses a sweet flavour with low total sugar content. These stunning features of purple carrot make it a good alternative to orange carrot. The colour of carrots results from the presence of pigmented compounds, for instance, carotenoids (α-carotene and β-carotene) in carrots produce orange colour, xanthophylls produce yellow colour, and anthocyanins produce purple colour; white carrots do not contain any colour pigments [8,9].

Carotenoids are highly valuable components of carrots. Moreover, their types alter with the chemical structure. For instance, carotenes such as α- and β-carotenes contain hydrocarbons, and xanthophylls such as β-cryptoxanthin, lutein, and zeaxanthin have an oxygen-containing functional group [10]. More importantly, carotenoids are vitamin A precursors, and vitamin A is crucial for eyesight and cell regulation [11]. Carotenoids also help to prevent cancer, bone-related diseases, cell oxidation, diabetes, obesity, and cardiovascular diseases [12,13].

The other compounds in carrot, especially in purple carrot, are anthocyanins, which are a subgroup of polyphenols and are well known for their free-radical scavenging activities [14]. Anthocyanins are produced against pathogens, UV radiation, pollination, and environmental stress during plant growth [15]. Additionally, polyphenolic compounds are useful for healthy human body functions, supporting protection against diabetes, cardiovascular diseases, osteoporosis, asthma, cancer, aging, and neuroprotection [16]. Anthocyanins are used as a food colourant to manufacture various food products such as canned strawberries, pasta, and rice muffins without egg or gluten [17,18].

In addition to colour pigments, carrot is a good source of vitamins B, C, E, and H; folic acid; pantothenic acid; and minerals such as K, Na, Ca, Mg, P, S, Mn, Fe, Cu, and Zn [19]. Moreover, carrot is rich in trace mineral molybdenum, which is crucial for the metabolism of carbohydrates and fats and iron absorption [20]. In addition, purple carrot contains 0 g fat, 31 g of carbohydrates with 1 g of fibre, and 1 g of protein; orange carrot includes 0.2 g of total fat, 6.9 g of carbohydrates with 2 g of fibre, and 0.7 g of protein.

Carrot contains simple sugars such as glucose, sucrose, and fructose and fibres such as cellulose and hemicelluloses. Moreover, carrot contains polyacetylenes, which might be able to destroy malignant cells such as leukaemia, myeloma, and lymphoma cells. Additionally, carrot contains luteolin that could protect against age-related symptoms in the brain, but the mechanisms are far from being elucidated [21,22].

Despite these interesting nutritional properties, many people do not consume carrot. Thus, to attract consumers, manufacturers prepare mixed carrot bags of yellow, purple, white, and orange carrots and call them “rainbow carrots”; moreover, mini (baby) carrots that are approximately up to 5 cm in size were created for consumption by young generations.

Therefore, the present study aimed to compare carrot varieties (different colours and sizes) in terms of bioactive contents and health-promoting properties. The present study also attempted to find the best carrot variety with high health benefits and elevated product quality for food processing. Thus far, such studies have not been conducted, especially in terms of determining polyphenolic and carotenoid contents and in vitro biological activities against enzymes related to diabetes (α-amylase and α-glucosidase), obesity (lipase), and age-related (acetylcholinesterase and butyrylcholinesterase) disorders. 

## 2. Materials and Methods

### 2.1. Chemicals

Standards of carotenoids, chlorophylls, and polyphenolics were purchased from Extrasynthese (Lyon, France). Acetonitrile, methanol, and formic acid for analyses of ultra-performance liquid chromatography (UPLC; gradient grade) and ascorbic acid were purchased from Merck (Darmstadt, Germany). Dipotassium hydrogen, orthophosphate dihydrogen, sodium phosphate monobasic, starch from potato, α-amylase from porcine pancreas (type VI-8; the European Community number (EC number) 3.2.1.1.; *p*-nitrophenyl-α-D-glucopyranoside, α-glucosidase from *Saccharomyces cerevisiae* (type I, EC number 3.2.1.20), lipase from porcine pancreas type II (EC number 3.1.1.3), *p*-nitrophenyl acetate, acetylcholinesterase from *Electrophorus electricus* (electric eel) (type VI-S; EC number 3.1.1.7), butyrylcholinesterase from equine serum (EC number 3.1.1.8), acetylthiocholine iodide, S-butyrylthiocholine chloride, and DTNB (5,5-dinitrobis-(2-nitrobenzoic acid)) were purchased from Sigma-Aldrich (Steinheim, Germany).

### 2.2. Plant Material and Sample Preparation

Normal and mini-sized carrots were purchased from Fusion Gusto (Dąbrowa, Poland). Micro carrots were purchased from Cato Produce (Johannesburg, South Africa) in June 2020.

Carrot samples were grouped according to their sizes and colours. “Normal size” carrots in diameter (d) were between 20 mm and 45 mm and they weighed (m) from 50 g to 150 g; “mini size” carrots—20 mm > d > 10 mm and 50 g > m > 8 g; “micro size” d < 10 mm and m < 8 g. 

Therefore, the following varieties of carrot were investigated: yellow carrot (micro (MYC), mini (MiYC), and normal (NYC), purple carrot (micro (MPC), mini (MiPC), and normal (NPC), orange carrot (micro (MOC), mini (MiOC), and normal (NOC), and white carrot (micro (MWC), mini (MiWC), and normal (NWC) (Figure 1). The carrot roots were washed, dried, cut into slices, and then frozen at −80 °C. The sliced carrots were then freeze-dried (24 h; Christ Alpha 1–4 LSC, Melsungen, Germany) and crushed by a laboratory mill (IKA A 11, Staufen, Germany) to obtain the homogeneous dry material for analysis.

### 2.3. Identification and Quantification of Polyphenols

The powder of roots (∼1 g) was mixed with 9 mL of the mixture containing HPLC-grade methanol: H_2_O (30:70%, *v*/*v*), ascorbic acid (2%), and acetic acid (1%) of the reagent. The extraction was performed twice by incubation for 20 min under sonication (Sonic 6D, Polsonic, Warsaw, Poland) and with periodic shaking [23]. Following, the slurry was centrifuged at 19,000× *g* for 10 min. The supernatant was filtered through a hydrophilic PTFE (polytetrafluoroethylene) 0.20 μm membrane (Millex Simplicity Filter, Merck, Darmstadt, Germany) and used for analysis. The extraction was performed in triplicate. Qualitative (liquid chromatography-quadrupole time-of-flight mass spectrometry (LC-MS-Q/TOF)) and quantitative (ultra performance liquid chromatography with photodiode array detector (UPLC-PDA)) analyses of polyphenols (phenolic acids at 320 nm, and anthocyanins at 520 nm) were performed according to Wojdyło et al. [24]. Polyphenols were separated by ACQUITY UPLC BEH C18 column (1.7 μm, 2.1 × 100 mm, Waters Corporation, Milford, USA) at 30 °C. The injection and elution of the samples (5 μL) were concluded in 15 min with a sequence of linear gradients and a flow rate of 0.42 mL/min. The solvent A (2.0% formic acid, *v*/*v*) and solvent B (100% acetonitrile) comprised the mobile phase. The procedure operated through gradient elution with 99–65% solvent A (0–12 min), solvent A was later lowered to 0% for condition column (12.5–13.5 min), and the gradient returned to the initial composition (99% A) for 15 min to re-equilibrate the column. In addition, in different varieties of carrot, the content of polymeric procyanidins was determined and performed according to Kennedy and Jones [25]. All estimations were performed in triplicate. The results were expressed as mg per kg of dry matter (dm).

### 2.4. Identification and Quantification of Carotenoids and Chlorophylls

To obtain the samples for the determination of carotenoids and chlorophylls, the lyophilized carrot powders (~0.20 g) containing 10% MgCO_3_ and 1% butylhydroxytoluene (BHT) were shaken with 5 mL of a ternary mixture of methanol/acetone/hexane (1:1:2, by vol.) at 300 rpm (DOS-10L Digital Orbital Shaker, Elmi Ltd., Riga, Latvia) for 30 min in the dark to prevent oxidation. The samples were centrifugated (4 °C, 7 min at 19,000× *g*; MPW- 350, Warsaw, Poland), and recovered supernatants were acquired after the 4 times re-extracted from solid residue. Combined fractions were evaporated. The pellet was solubilized using methanol and filtered through a hydrophilic polytetrafluoroethylene (PTFE) 0.20 μm membrane (Millex Samplicity^®^ Filter, Merck, Darmstadt, Germany). Then carotenoids and chlorophylls were analysed by LC-MS-Q/TOF (identification) and UPLC-PDA (quantification) on an ACQUITY UPLC BEH RP C18 column protected by a guard column of the same materials (1.7 mm, 2.1 mm × 100 mm, Waters Corp., Milford, MA, USA) was performed at 30 °C. The elution solvents were a linear gradient of acetonitrile:methanol (70:30%, *v*/*v*) (A) and 0.1% formic acid (B) at flow rates of 0.42 mL/min. The analysis was performed at 450 nm (carotenoids) and 660 nm (chlorophylls). The obtained spectra and retention times were compared with the authentic standards and in this way the bioactive compounds were determined. The tests were implemented in triplicate, and the results are presented as mg per kg of dm. 

### 2.5. Determination of Biological Activities of Carrot Varieties

Roots were investigated for α-amylase, α-glucosidase [26], and lipase inhibitory effects [27]. The acarbose was used as a positive control in the case of α-amylase and α-glucosidase, whereas orlistat was used as a positive control for pancreatic lipase. Additionally, the activities of cholinesterase inhibitors were evaluated with the acetylcholinesterase (AChE) and butyrylcholinesterase (BuChe) methods according to Ferreres et al. [28]. 

The results were presented as IC_50_ (mg/mL) (the amount of sample can reduce enzyme activity by 50%). All tests: α-amylase, α-glucosidase, anti-lipase, and anti-cholinergic activity were performed in triplicate using a SynergyTM H1 microplate reader (BioTek, Winooski, VT, USA).

### 2.6. Statistical Analysis

The two-way analysis of variance (ANOVA, *p* ≤ 0.05) and Duncan’s test were performed by Statistica version 13.3 (Stat-Soft, Cracow, Poland). The results are demonstrated as the mean value (*n* = 3).

## 3. Results and Discussion

### 3.1. Identification and Quantification of Phenolic Compounds in Coloured Carrot Varieties

Polyphenolic compounds are the secondary plant metabolites and are crucial for the colour, nutritional, antioxidant, and sensory features of foods. In the present study, the polyphenolic compositions were determined by UPLC-PDA-Q/TOF-MS, and the results are shown in Table 1. Twenty-four phenolic compounds (15 phenolic acids and nine anthocyanins) were detected. The identified phenolic acids showed similarities for purple, yellow, orange, and white carrot varieties to those reported in the literature [29]. The determined phenolic acids were derivatives of caffeic, caffeoylquinic, coumaroylquinic, ferulic, and feruloylquinic acid. In the present study, 3-O-caffeoylquinic acid, 3-O-feruloylquinic acid, O-q-coumaroylquinic acid, dicaffeoylquinic acid derivative, and di-ferulic acid derivative were detected in white and orange carrots for the first time. 4-O-caffeoylquinic acid, 3-O-feruloylquinic acid, di-ferulic acid derivative, and diferuoylquinic acid derivative were found in yellow carrots. 4-O-caffeoylquinic acid, 3-O-feruloylquinic acid, O-q-coumaroylquinic acid, dicaffeoylquinic acid derivative, di-ferulic acid derivative, and diferuoylquinic acid derivative were detected in purple carrots for the first time.

Table 2 presents the contents of phenolic compounds in carrot samples. The carrot samples showed significant differences in polyphenolic compositions (*p* ≤ 0.05).

In the present study, di-ferulic acid derivative and 5-O-caffeoylquinic acid (5-CQA) were found in all coloured carrot samples. In the literature, 5-CQA (chlorogenic acid) was also detected in sea fennel, bean, and spinach, and it is important because of health benefits such as anti-carcinogenic, anti-inflammatory, anti-diabetic, and anti-obesity properties [30,31]. Other phenolic acids were found in different amounts in all carrot varieties. The highest total phenolic acid contents were observed in NPC (945.00 mg/100 g dm), MiOC (503.93 mg/100 g dm), and MiPC (479.33 mg/100 g dm). According to the obtained results, mini-sized and purple carrot samples had abundant phenolic acid content. Thus, the differences in phenolic acid profiles of carrots might be related to the root size, colour, and chemical compositions of the carrot varieties. Orange, purple, yellow, and white carrots are results of phenolic and carotenoid contents during the maturation period as well as storage temperature [32,33].

Nine anthocyanins were also identified in purple carrot samples of each size (Table 1). The identified anthocyanins showed similar compounds as those reported in the literature [34,35,36,37]. Five of the anthocyanidins were quantified in purple carrot varieties. In the present study, NPC and MiPC showed the presence of five anthocyanins (cyanidin-3-O-xylosyl-glucosylgalactoside, cyanidin-3-O-xylosyl-galactoside, cyanidin-3-O-xylosyl-cinpoyl-glucosylgalactoside, cyanidin-3-O-xylosyl-feruloyl-glucosylgalactoside, and cyanidin-3-O-xylosyl-p-coumaroylglucosyl-galactoside). However, MPC showed only two anthocyanins (cyanidin-3-O-xylosyl-galactoside and cyanidin-3-O-xylosyl-feruloyl-glucosylgalactoside). Comparison of the total anthocyanin contents of purple carrots showed that NPC had the highest anthocyanin content (378.48 mg/100 g dm), followed by MiPC (255.08 mg/100 g dm) and MPC (8.21 mg/100 g dm).

Procyanidins are subclasses of proanthocyanidins and are known for their anti-hypertensive, anti-allergic, antioxidant, and anti-microbial activities. Procyanidins have not been quantified in carrot varieties thus far. The present study is the first to quantify procyanidin contents in carrots. In the present study, the highest procyanidin contents were found in MiWC, NOC, and NWC. Thus, normal-sized and white carrot samples were rich in polymeric procyanidins. The structures of procyanidins consist of stereochemistry, hydroxylation pattern, flavan-3-ol constitutive units, and degree of polymerisation (DP, number of flavanol units) [38]. In the present study, the highest DP values were detected in MiWC (2.06), NYC (1.99), and MiYC (1.72). According to the DP values noted in the present study, mini-sized and yellow carrot samples were rich in flavanol units. Therefore, MiWC showed high procyanidin contents and DP values.

To summarise the polyphenolic contents of coloured carrot varieties, the highest total phenolic contents were found in NPC, MiPC, and MiOC. Thus, purple carrot has the highest total content of polyphenols, which was also confirmed by other authors [29,36,39,40,41]. Moreover, phenolic contents may change depending on genes, environment, and climate factors [42].

### 3.2. Identification and Quantification of Carotenoids in Carrot Varieties

Table 3 shows the content of carotenoids and chlorophylls of the 12 carrot varieties with significant differences (*p* ≤ 0.05). In the present study, violaxanthin, astaxanthin, lutein, zeaxathin, α-cryptoxanthin, β-cryptoxanthin, (6R)-δ-carotene, α-carotene, γ-carotene, ε-carotene, β-carotene, and trans-apo-carotenal were identified as carotenoids. NPC exhibited the highest ratio of different carotenoid types (lutein, zeaxathin, α-cryptoxanthin, β-cryptoxanthin, γ-carotene, and trans-apo-carotenal). Trans-apo-carotenal was detected only in NPC.

Carotenoids, which cannot be synthesised by animals and humans, have to be obtained from the diet as crucial precursors for healthy body functions [43]. According to Alasalvar et al. [39], β-carotene and α-carotene are the most abundant carotenoids in carrots. In the present study, β-carotene was detected in NYC > MYC > MiYC > MPC > NWC, whereas α-carotene was observed in NYC > MiOC > NOC > MiYC = NWC > MiWC. The highest content of β-carotene was found in NYC and MYC, whereas the highest α-carotene content was observed in NYC and MiOC. Thus, NYC showed the highest presence of both α and β-carotenes. Reif et al. [44] showed a different trend in their study and indicated a low amount of β-carotene in yellow carrots. The contents of β-carotene and α-carotene change with temperature, which may have caused discrepancies in the observed results.

Carotenoids are unstable pigments, which can create some problems while working with them. For instance, orange carrot is known to be a rich source of β-carotene and α-carotene [44]. However, in the present study, α-carotene was not detected in MOC; moreover, according to the applied method, and retention time differences in β-carotene were not found in NOC, MiOC, or MOC. This was a surprising observation but was confirmed by the standard of β-carotene. Maybe the content was very low, and therefore no peak of this compound was observed. Moreover, α- and β-carotene can be pressed by chlorophylls and xanthophylls as well [45]. In addition, storage period and temperature during the storage are other important factors for pigmentation.

High levels of δ-carotene and ε-carotenes were found in orange carrot samples of various sizes and have not been reported before.

Six xanthophyll pigments were identified and quantified in carrot samples (Table 3). In the present study, lutein was found solely in NPC and NYC. Similar results have been reported in the literature [44,46].

Chlorophyll pigments are essential to prevent chronic diseases [47]. In the present study, pheophorbide a, chlorophylls a, and b were found as chlorophylls pigments in the carrot samples. Pheophorbide a was solely determined in NOC and NPC, in turn chlorophyll a was identified in NOC, NYC, NWC, MiYC, MWC, and MYC. Except for MiPC, all carrot varieties showed the content of chlorophyll b. Moreover, chlorophyll a and b contents were high in NYC. Thus, chlorophyll contents were found to change within cultivars [48].

To summarise, the total carotenoid contents were the highest in NYC, MiOC, NPC, and MPC. Normal-sized and purple carrot samples showed greater total carotenoid contents. The total chlorophyll contents were the highest in NYC, MiYC, NPC, and MYC. Therefore, normal-sized and yellow carrot samples were rich in chlorophyll contents. The highest total carotenoid and chlorophyll results were observed in NYC, MiOC, NPC, and MPC. Normal-sized and purple carrot samples were rich in carotenoid and chlorophyll contents. The carotenoid contents of carrots may change depending on variety, maturity, growth conditions, growing season, soil, and genetic factors [49].

### 3.3. Enzyme Inhibitory Activities by Coloured Carrot Roots

The enzyme inhibitory activities of 12 carrot varieties against α-amylase, α-glucosidase, pancreatic lipase, acetylcholinesterase (AChE), and butyrylcholinesterase (BuChE) were evaluated by in vitro assays and expressed as IC_50_—the amount of sample can reduce enzyme activity by 50%. The results are shown in Table 4. The carrots showed significant differences in all five inhibition activities (*p* ≤ 0.05). It should be noted that enzyme inhibition activities of carrot varieties have not been investigated thus far.

The 12 carrot varieties were assayed for their anti-diabetic properties (inhibition of α-amylase and α-glucosidase enzymes). The enzymes α-amylase and α-glucosidase are responsible for carbohydrate digestion, and the inhibition of these enzymes may decrease postprandial blood glucose levels by reducing the breakdown of polysaccharides into glucose. The bioactivities of many plant species have been shown to decrease early-stage diabetes-related enzyme activities [50]. In the present study, the IC_50_ values of the carrot samples for inhibiting α-amylase ranged from 107.85 to 807.92 mg/mL. MiWC and MOC showed the highest activities against α-amylase, whereas the lowest activity was shown by NOC. Most carrot extracts had higher IC_50_ for α-amylase than sour cherry, red grapefruit, pineapple, orange, and kiwi [27]. To clarify, α-amylase inhibitory activity is a result of bioactive compounds of plants such as glycosides, polysaccharides, steroids, and terpenoids [51]. Overall, mini carrots showed the highest inhibitory effect against α-amylase.

The IC_50_ values of the carrot samples for inhibiting α-glucosidase ranged from 97.02 to 897.79 mg/mL. MiPC and MYC showed the highest inhibitory activities against α-glucosidase, whereas the lowest result was shown by MiWC. The IC_50_ values for α-glucosidase were higher than chokeberry, apple, pear, and blackberry, a group of fruits recognized as important in the prevention of diabetes [27]. A previous study reported that the high anti-diabetic activity of onion is associated with its content of phenolic acids, flavonoids, and anthocyanins [52]. Polyphenol and carotenoid contents were shown to increase the anti-diabetic activities of fruits and vegetables [53,54]. A comparison of α-amylase and α-glucosidase inhibitory activities of carrot samples revealed that MiWC had the highest α-amylase inhibitory activity but the lowest α-glucosidase inhibitory activity. Micro-sized samples showed the highest α-glucosidase activities. Moreover, α-amylase inhibitory activity demonstrated non-significant correlations with total carotenoid and chlorophyll contents (R^2^ = 0.05), phenolic acid content (R^2^ = 0.04), and procyanidin content (R^2^ = 0.01). However, chemical contents of carrot varieties showed correlations with the α-glucosidase inhibitory activity that were different from those observed for α-amylase inhibitory activity; the following R^2^ values were noted for procyanidin, and total carotenoid and chlorophyll contents: 0.50 and 0.01, respectively, and no correlation was observed with phenolic acid content (R^2^ = 0.00). Therefore, phenolics are correlated with diabetes-related enzymes [55]; however, in the present study high positive correlation in carrot varieties was not determined.

Pancreatic lipase hydrolyses triacylglycerols into free fatty acids, bile salts, and fat-soluble vitamins [56]. In the present study, the IC_50_ values for pancreatic lipase inhibitory activities ranged from 5.29 to 12.25 mg/mL. The highest IC_50_ values were observed for NOC (5.29), MiYC (5.69), MYC (6.05), and NPC (6.12); however, NWC (12.25) and MWC (11.36) showed the lowest activities against pancreatic lipase. According to Fabroni [57], total anthocyanin content correlates with pancreatic lipase activity; however, in the present study, similar results were not observed, because normal-sized orange carrot exhibited elevated activities against pancreatic lipase. Moreover, in the literature lentil cultivars have shown potent activity (IC_50_ from 6.26 to 9.26 mg/mL) against pancreatic lipase [58]. Hence, phenolics, ascorbic acid, and carotenoids are responsible for pancreatic lipase inhibition [59].

More importantly, the carrot samples exhibited higher inhibitory activities against pancreatic lipase than against diabetes-related enzymes. The pancreatic lipase inhibitory activity showed a positive correlation with the total carotenoid and chlorophyll contents (R^2^ = 0.29), phenolic acid content (R^2^ = 0.17), and to a low extent with procyanidin content (R^2^ = 0.06).

The inhibitions of AChE and BuChE are thought to be important for the diagnosis and treatment of diseases such as bladder distention, glaucoma, myasthenia gravis, and Alzheimer’s disease [60]. In the present study, carrot samples were investigated for potential inhibitor activities against AChE and BuChE. The IC_50_ values of the carrot samples for AChE inhibition ranged from 10.14 to 18.96 mg/mL. The highest AChE inhibitory activities were observed for MPC (10.14 mg/mL), MWC (12.05 mg/mL), and MiWC (12.31 mg/mL), whereas the lowest result was observed for NPC (18.96 mg/mL). Moreover, in the literature white ginger exhibited more potent activities for AChE inhibition than red ginger variety [61]. In addition, a previous study showed that the IC_50_ value of Hippophaë cultivars for inhibiting AChE ranged from 20.16 to 40.60 mg/mL [62]. *Dipsacus* root showed higher activity against AChE than the other parts of the plant [63]. The content of polyphenols and carotenoids in carrots was shown to correlate with activities of cholinesterase inhibitors [64,65]. In the present study, micro-sized and white carrot varieties showed elevated activities against AChE. Furthermore, the IC_50_ values of the carrot samples for inhibiting BuChE ranged from 7.83 to 19.02 mg/mL. MPC (7.83 mg/mL), NPC (7.85 mg/mL), and NYC (8.01 mg/mL) showed the highest BuChE inhibitory activities, whereas the lowest activity was shown by NWC (19.02 mg/mL). Thus, normal-sized and purple carrot samples showed superior activities against BuChE. On the other hand, AChE inhibitory activity was correlated with the content of total carotenoid and chlorophyll (R^2^ = 0.27), procyanidins (R^2^ = 0.17), and phenolic acids (R^2^ = 0.02), whereas BuChE inhibitory activity was correlated with the content of procyanidins (R^2^ = 0.17), total carotenoid and chlorophyll (R^2^ = 0.07), and phenolic acids (R^2^ = 0.03). Therefore, phenolics are thought to be related to cholinesterase inhibition [66].

Overall, MiPC showed the highest activities against α-glucosidase, whereas MPC exhibited the highest activity against AChE and BuChE. NOC showed the highest inhibition activity against pancreatic lipase. Thus, purple carrot samples were more functional for enzyme inhibition than other coloured carrots of different sizes.

## 4. Conclusions

The present study evaluated 12 carrot varieties of different sizes and colours for bioactive compounds and their biological activities. The study results showed that purple carrot samples had the highest values for the content of polyphenolics and carotenoids, with the highest activities against cholinesterase. Normal yellow carrot showed the lowest values for the content of polyphenols. Micro white carrot demonstrated the lowest results for total phenolic acid, total carotenoid, and chlorophyll content. Moreover, the activities against diabetes-, obesity- and aging-related enzymes of the carrot varieties were found to be correlated with the content of phenolics and carotenoids.

Overall, the present study attempted to find the best carrot variety with high nutrients for food processing. Normal purple carrot showed the highest health-promoting activities in all tests, followed by mini purple carrot. Thus, different-sized purple carrot varieties can provide high contents of bioactive compounds to combat oxidative stress-related diseases.

However, the mechanisms of the pro-health action of carrots were not confirmed in this study, and their activities as α-amylase, α-glucosidase, lipase, acetylocholinesterase, and butyrylocholinesterase inhibitors was rather poor. Probably the mechanisms of their action are more complex, and the possible health-promoting effect results from the synergy of many compounds, including fibre, phytochemicals, vitamins, and minerals. Therefore, it would be worth continuing research on different varieties of carrots.

## Figures and Tables

**Figure 1 foods-10-00808-f001:**
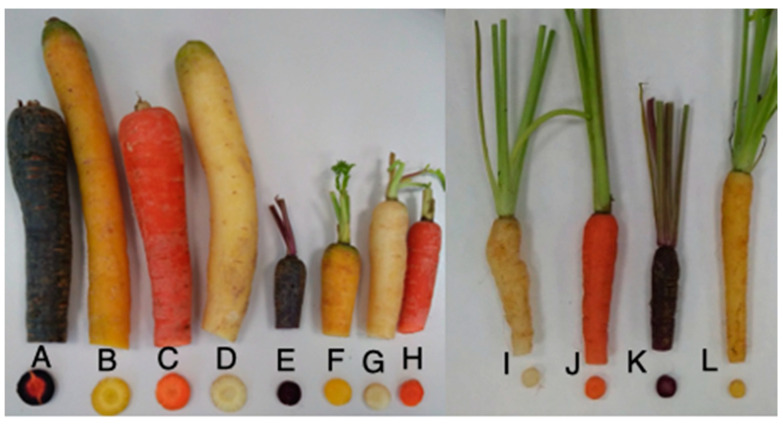
Types of carrots and varieties (A—normal purple carrot (NPC), B—normal yellow carrot (NYC), C—normal orange carrot (NOC), D—normal white carrot (NWC), E—mini purple carrot (MiPC), F—mini yellow carrot (MiYC), G—mini white carrot (MiWC), H—mini orange carrot (MiOC), I—micro white carrot (MWC), J—micro orange carrot (MOC), K—micro purple carrot (MPC), L—micro yellow carrot (MYC)).

**Table 1 foods-10-00808-t001:** Overall phenolic compounds identified in 12 carrots varieties.

Compound	R_t_ (min)	Λmax (nm)	MS [M-H] (*m*/*z*) *	MS/MS (*m*/*z*)
Phenolic acids				
3-O-caffeoylquinic acid	5.90	324	353	135/179/191
Caffeic acid-hexoside	6.31–6.82–7.45	324	341	179/135
5-O-caffeoylquinic acid	7.54	325	353	179/191
4-O-caffeoylquinic acid	8.03	325	353	179/191
Ferulic acid-hexoside	8.69–9.22–9.80	324–325	355	193/175
Ferulic acid di-hexoside	9.41	324	517	355/193/175
3-O-feruloylquinic acid	10.10	322	367	173/193
O-q-coumaroylquinic acid	10.70	312	337	191
4-O-feruloylquinic acid	10.84	323	367	173/193
Caffeic acid-hexoside	10.90	324	341	179/135
5-O-feruloylquinic acid	11.23	325	367	191/193
Dicaffeoylquinic acid derivative	13.68–14.58–15.11	327	515	185/353
Di-ferulic acid derivative	14.27	327	527	203/365/366
Ferulic acid	14.88	324	193	
Diferuoylquinic acid derivative	16.20	324	543	
Anthocyanins				
Cyanidin-3-O-xylosyl-glucosylgalactoside	5.58	517	743	287
Delphinidin-3-O-rutinoside	5.86		611	303
Cyanidin-3-O-xylosyl-galactoside	6.14	518	581	287
Delphinidin-3-O-sambubioside	6.90		597	303
Cyanidin-3-O-xylosyl-sinapoyl-glucosylgalactoside	7.22	530	949	287
Cyanidin-3-O-xylosyl-feruloyl-glucosylgalactoside	7.57	528	919	287
Cyanidin-3-O-xylosyl-p-coumaroylglucosyl-galactoside	7.68	527	889	287
Ferulic acid derivative of pelargonidin 3-xylosylglucosylgalactoside	8.22	527	903	271
Ferulic acid derivative of peonidin 3-xylosylglucosylgalactoside	8.34	530	933	301

* MS [M + H]^+^—for anthocyanins.

**Table 2 foods-10-00808-t002:** Quantifications of phenolic compounds in 12 carrot varieties.

Compounds	Normal Orange	Normal Purple	Normal Yellow	Normal White	Mini Orange	Mini White	Mini Yellow	Mini Purple	Micro White	Micro Yellow	Micro Orange	Micro Purple
3-O-caffeoylquinic acid	5.06 ^‡^ f	46.55 a	nd k	3.24 g	12.51 c	7.84 e	nd k	15.52 b	1.05 j	nd k	12.08 d	1.94 h
5-O-caffeoylquinic acid	32.48 g	534.58 a	16.37 l	17.09 k	322.54 b	19.35 j	32.73 f	289.94 c	3.98 m	72.75 d	34.70 e	21.71 h
4-O-caffeoylquinic acid	nd f	55.24 a	0.40 e	nd f	nd f	nd f	4.22 d	10.56 c	nd f	20.04 b	nd f	nd f
Ferulic acid-hexoside	7.92 e	94.73 a	nd h	nd h	9.65 d	3.89 f	nd h	67.18 b	0.87 g	nd h	nd h	12.04 c
Ferulic acid di-hexoside	nd e	54.16 a	nd e	nd e	nd e	nd e	nd e	8.63 b	nd e	nd e	6.64 c	3.11 d
3-O-feruloylquinic acid	2.79 e	52.87 a	2.08 f	3.33 d	5.33 b	4.39 c	1.87 g	nd k	0.72 j	nd k	nd k	1.62 h
O-q-coumaroylquinic acid	0.83 g	22.20 b	nd h	1.45 f	1.63 e	2.51 d	nd h	27.64 a	nd h	nd h	nd h	5.95 c
5-O-feruloylquinic acid	nd h	7.80 a	0.96 f	nd h	nd h	0.93 g	1.70 d	4.53 c	nd h	5.77 b	nd h	1.13 e
Dicaffeoylquinic acid derivative	8.21 c	29.27 a	nd h	3.10 f	nd h	10.14 b	nd h	nd h	2.65 g	nd h	3.57 d	3.51 e
Di-ferulic acid derivative	80.49 b	7.08 k	9.53 j	11.78 h	75.13 d	27.54 g	31.46 e	27.85 f	0.92 m	204.36 a	79.42 c	6.46 l
Ferulic acid	30.41 e	33.73 d	1.04 k	nd m	42.69 b	16.83 g	12.18 h	27.50 f	0.92 l	62.14 a	34.24 c	5.31 j
Diferuoylquinic acid derivative	nd f	6.79 b	0.77 e	nd f	nd f	nd f	2.31 d	nd f	nd f	10.42 a	nd f	2.98 c
4-O-feruloylquinic acid	13.35 f	nd k	7.74 g	5.78 h	34.45 c	14.25 e	18.31 d	nd k	0.87 j	76.72 a	36.94 b	nd k
Caffeic acid-hexoside	1.00 b	nd c	nd c	nd c	nd c	nd c	nd c	nd c	nd c	5.71 a	nd c	nd c
Total phenolic acids	182.54 f	945.00 a	38.90 l	45.77 k	503.93 b	107.68 g	104.78 h	479.33 c	11.98 m	457.92 d	207.59 e	65.76 j
Cyanidin-3-O-xylosyl-glucosylgalactoside		45.04 a						22.14 b				nd c
Cyanidin-3-O-xylosyl-galactoside		16.01 b						51.88 a				0.75 c
Cyanidin-3-O-xylosyl-cinpoyl-glucosylgalactoside		43.09 a						8.83 b				nd c
Cyanidin-3-O-xylosyl-feruloyl-glucosylgalactoside		257.47 a						168.65 b				7.46 c
Cyanidin-3-O-xylosyl-p-coumaroylglucosyl-galactoside		16.87 a						3.60 b				nd c
Total anthocyanins		378.48 a						255.08 b				8.21 c
Polymeric procyanidins	69.62 b	44.44 j	20.64 m	53.80 c	38.05 l	78.92 a	51.19 f	46.50 h	52.93 e	49.26 g	44.09 k	53.05 d
DP	1.16 j	1.31 h	1.99 b	1.44 f	1.00 k	2.06 a	1.72 c	1.62 d	1.49 e	1.48 e	1.35 g	1.32 gh
Total Polyphenolic Content	253.32 e	1369.23 a	61.53 m	101.01 k	542.98 c	188.66 g	157.69 h	782.53 b	66.40 l	508.66 d	253.03 f	128.34 j

nd—not detected; polyphenols, polymeric procyanidins—mg/100 g dm; DP—degree of polymerisation; significant at *p* ≤ 0.05; ‡ values (mean of three replications) followed by the same letter within the same column were not significantly different (*p* > 0.05) according to Duncan’s test.

**Table 3 foods-10-00808-t003:** Quantifications of carotenoids and chlorophylls of 12 carrot varieties.

Compounds	Normal Orange	Normal Purple	Normal Yellow	Normal White	Mini Orange	Mini White	Mini Yellow	Mini Purple	Micro White	Micro Yellow	Micro Orange	Micro Purple
Violaxanthin	0.03 ^‡^ abc	0.04 abc	0.05 ab	0.01 c	0.02 bc	0.01 c	0.06 a	0.04 abc	0.01 c	0.04 abc	0.02 bc	0.03 abc
Astaxanthin	2.58 h	3.43 f	5.08 c	0.32 k	2.61 h	0.32 k	8.09 a	4.65 d	0.41 j	6.78 b	3.01 g	4.47 e
Lutein	nd c	0.19 a	0.14 b	nd c	nd c	nd c	nd c	nd c	nd c	nd c	nd c	nd c
Zeaxanthin	0.17 b	0.70 a	nd c	nd c	nd c	nd c	nd c	nd c	nd c	nd c	nd c	nd c
α-cryptoxanthin	0.02 b	0.05 a	nd b	nd b	0.01 b	nd b	nd b	nd b	nd b	nd b	0.01 b	0.02 b
Beta-cryptoxanthin	nd f	1.65 a	0.27 b	0.08 e	nd f	0.10 e	0.20 c	nd f	0.09 e	0.27 b	nd f	0.13 d
(6R)-δ-carotene	0.47 a	0.24 d	0.05 fgh	0.08 f	0.03 h	0.06 fgh	0.47 a	0.07 fg	0.17 e	0.41 b	0.04 gh	0.30 c
α-carotene	6.91 c	nd f	14.43 a	2.88 d	13.28 b	1.40 e	2.89 d	nd f	nd f	nd f	nd f	nd f
γ-carotene	4.02 d	12.22 a	0.61 g	0.60 g	5.25 c	0.41 h	nd k	nd k	0.26 j	2.09 f	2.63 e	11.79 b
ε-carotene	0.16 f	0.54 b	1.02 a	0.16 f	0.30 e	nd h	0.40 d	nd h	0.10 g	0.46 c	nd h	0.46 c
β-carotene	nd f	nd f	14.49 a	0.85 e	nd f	nd f	3.22 c	nd f	nd f	4.14 b	nd f	1.71 d
Trans-apo-carotenal	nd a	0.01 a	nd a	nd a	nd a	nd a	nd a	nd a	nd a	nd a	nd a	nd a
Total carotenoids	14.36 f	19.07 c	36.14 a	4.98 j	21.50 b	2.30 l	15.33 e	4.76 k	1.04 m	14.19 g	5.71 h	18.91 d
Pheophorbide a	0.01 ab	0.02 a	nd b	nd b	nd b	nd b	nd b	nd b	nd b	nd b	nd b	nd b
Chlorophyll a	0.03 c	nd d	0.15 a	0.01 cd	nd d	nd d	0.06 b	nd d	0.01 cd	0.06 b	nd d	nd d
Chlorophyll b	0.01 d	0.08 b	0.16 a	0.05 bc	0.01 d	0.02 cd	0.07 b	nd d	0.01 d	0.01 d	0.03 cd	0.01 d
Total chlorophylls	0.05 def	0.10 c	0.31 a	0.06 de	0.01 g	0.02 fg	0.13 b	0.00 g	0.02 fg	0.07 d	0.03 efg	0.01 g
Total carotenoid and chlorophyll content	14.42 f	19.16 c	36.46 a	5.04 j	21.50 b	2.33 l	15.45 e	4.76 k	1.05 m	14.25 g	5.75 h	18.91 d

nd—not detected; carotenoids and chlorophylls—mg/100 g dm; significant at *p* ≤ 0.05; ‡ values (mean of three replications) followed by the same letter within the same column were not significantly different (*p* > 0.05) according to Duncan’s test.

**Table 4 foods-10-00808-t004:** In vitro inhibition activity (α-amylase, α-glucosidase, pancreatic lipase, acetylcholinesterase, butyrylcholinesterase (IC_50_, mg/mL) of different carrot varieties.

Type of Carrot	α-Amylase *	α-Glucosidase **	Lipase ***	AChE ^1^	BuChe ^2^
Normal orange	807.92 ^‡^ a	125.93 h	5.29 l	13.13 h	9.36 h
Normal yellow	441.45 b	151.08 f	6.83 e	14.14 g	8.01 k
Normal purple	239.49 c	643.91 b	6.12 h	18.96 a	7.85 l
Normal white	219.21 d	116.53 j	12.25 a	16.05 c	19.02 a
Mini orange	128.96 h	307.09 d	6.81 e	16.45 b	15.77 c
Mini yellow	122.58 k	356.68 c	5.69 k	15.08 e	13.94 e
Mini purple	127.06 j	97.02 m	6.94 d	15.74 d	16.8 b
Mini white	107.85 m	897.79 a	9.58 c	12.31 j	14.03 d
Micro orange	111.03 l	251.83 e	6.29 g	14.61 f	9.03 j
Micro yellow	218.66 e	104.18 l	6.05 j	14.61 f	10.75 g
Micro purple	181.37 g	148.96 g	6.49 f	10.14 l	7.83 l
Micro white	199.79 f	106.06 k	11.36 b	12.05 k	11.05 f

Significant at *p* ≤ 0.05; ‡ values (mean of three replications) followed by the same letter within the same column were not significantly different (*p* > 0.05) according to Duncan’s test, * inhibition activity of acarbose < 5 mg/mL; ** inhibition activity of acarbose < 5 mg/mL; *** inhibition activity of Orlistat < 1 mg/mL; ^1^ AChE–acetylcholinesterase; ^2^ BuChe –butyrylcholinesterase.

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
