# Peer review of "Nutritional, Phytochemical Characteristics and In Vitro Effect on α-Amylase, α-Glucosidase, Lipase, and Cholinesterase Activities of 12 Coloured Carrot Varieties"

_foods, 2021, doi:10.3390/foods10040808_

Round 1

Reviewer 1 Report

This work must be rewritten and refocused on what was really acomplished and on the meaningful results obtained.

  • Starting with the title, which should really be: “Nutritional and Phytochemical Characteristics of 12 Coloured Carrot Varieties”. No “glycolipid digestion” was assessed, only the inhibition of α-amylase and α-glucosidase (which break down starch and disaccharides to glucose) and lipase (lipid hydrolysis). Glycolipids are a group of molecules not to be confused with a mixture containing both carbohydrates and lipids.
  • In the abstract the authors refer that “the highest antioxidant activities (measured by ORAC) were determined in mini and normal purple carrots.”, however no antioxidant results are presented in this manuscript, so this should be removed from the Abstract. In addition, I do not agree that the results of inhibition against diabetes-related enzymes (not “diabetic enzymes” , which does not exist) was high, since the authors used carrot extract concentrations which were so high that the IC50 values obtained correspond in fact to what should be considered as “not active”.
  • The authors shoud define better what is meant by “normal, mini-sized and micro-carrots”.  Although we understand the difference in relative scale,  a more precise notion of the sizes is needed (e.g., average weight / carrot in each group)
  • Lines 70-73: the enzymes used in the assays must be properly characterized, it is not enough to refer that “the additional reagents were purchased from Sigma-Aldrich (Steinheim, Germany).”. For instance, acetylcholinesterases differ according to the source, and therefore inhibition also changes if you’re using the Eletrophorus electricus enzyme, the human enzyme, or extracted form an invertebrate. This information is lacking (here and in the section 2.5, where bioactivities should be properly described and referenced). The complete information (E.C. classification, origin of the enzyme, Sigma-Aldrich reference) is mandatory.
  • Line 111: Write “The pellet was solubilized” instead of “The pellet was subtilized”
  • Line 125: the authors mention antioxidant assays (ABTS, ORAC, FRAP), but not the methods and references; actually, no antioxidant results are presented either, so why mention this activity ?
  • Lines 121-127: The use of positive controls for the bioactivities is essential, since it allows the comparison of the results obtained for the samples with standard molecules. The authors do not mention these controls (and indeed there are no results for controls is Table 4). In addition, the concentrations used for the assays (in the order of mg/mL) are too high to be of interest.
  • Line 216: the caption should be “3.3. Inhibitory activities of digestive enzymes by coloured carrot roots” instead of “3.3. Inhibitory activities of coloured carrot roots through digestive enzymes”
  • Line 219: acetylcholinesterase (AChE), and butyrylcholinesterase (BuChE) are not digestive enzymes, so they should not be included under “digestive enzymes” ; a suggestion is using a different caption in 3.3, such as “Enzyme inhibitory activities by coloured carrot roots”, since that would include all the enzymes studied.
  • Lines 219-221: IC50, as correctly defined in Materials and Methods (line 125), is not “the lowest concentration of carrot that showed the highest inhibition potential”, but the concentration that inhibits activity by 50%. This must be corrected.
  • Where are the results of antioxidant activities (ABTS, ORAC, FRAP) referred in Materials and Methods?
  • The enzyme inhibitory activities (table 4) are too high to be of interest, the IC50 values presented in this work mean that all the samples should be considered as having no activity. In our lab we never go above 150 µg/mL, i.e., 0.150 mg/mL. Concentrations higher than that should not even be tested just to find an effect.

I suggest the authors rewrite this work focusing only on the phytochemical and nutrient characterization of the carrot varieties, which is already interesting and useful, besides carrying out the necessary corrections.

Author Response

Thank you for your review. Below are presented the answers to your suggestions

General comment: This work must be rewritten and refocused on what was really acomplished and on the meaningful results obtained.

Answer: According to your suggestion the results have been reanalyzed and the work has been somewhat rewritten. Below are answers to your suggestions and comments.

Comment: Starting with the title, which should really be: “Nutritional and Phytochemical Characteristics of 12 Coloured Carrot Varieties”. No “glycolipid digestion” was assessed, only the inhibition of α-amylase and α-glucosidase (which break down starch and disaccharides to glucose) and lipase (lipid hydrolysis). Glycolipids are a group of molecules not to be confused with a mixture containing both carbohydrates and lipids.

Answer: Your comment is very valuable. In connection with the above, the new title proposed by us is - Nutritional, Phytochemical Characteristics and In Vitro Effect on α-Amylase, α-Glucosidase, Lipase, and Cholinesterase Activities 12 Coloured Carrot Varieties. In our opinion, this title fully describes the results presented in this manuscript.

Comment: In the abstract the authors refer that “the highest antioxidant activities (measured by ORAC) were determined in mini and normal purple carrots.”, however, no antioxidant results are presented in this manuscript, so this should be removed from the Abstract.

Answer: It has been removed. The results of the antioxidant activity were removed during the editing of the final version of manuscript. Information on this aspect in the abstract and purpose is due to the inattention of the authors and have been removed.

Comment: In addition, I do not agree that the results of inhibition against diabetes-related enzymes (not “diabetic enzymes” , which does not exist) was high, since the authors used carrot extract concentrations which were so high that the IC50 values obtained correspond in fact to what should be considered as “not active”.

Answer: According to diabetes-related enzymes, the study results have been shown for the present study to compare with carrot varieties. In addition, we cannot agree with the statement that the obtained IC50 for carrot extract are so high that should be considered as “not active”. Most of carrot extracts have higher IC50 for α-amylase, than sour cherry, red grapefruit, pineapple, orange or kiwi. Also IC50 for α-glucosidase are higher than for chokeberry, apple, pear or blackberry, a group of fruits recognized as important in the prevention of diabetes (Podsędek et al., 2014). Obviously, these results are much lower than in the case of isolated compounds or pharmaceutical preparations. However, they do provide some important screen regarding bioactivity properties of carrots.

Comment: The authors should define better what is meant by “normal, mini-sized and micro-carrots”.  Although we understand the difference in relative scale,  a more precise notion of the sizes is needed (e.g., average weight / carrot in each group).

Answer: Better definition has been provided on the revised manuscript. Carrot samples were grouped according to their sizes and colours. “Normal size” carrot in diameter (d) were between 20 mm and 45 mm and they weighed (m) from 50 g to 150 g; “mini size” carrots – 20 mm>d>10 mm and 50 g >m> 8 g;   “micro size”- d< 10 mm and m< 8 g”.

Comment: Lines 70-73: the enzymes used in the assays must be properly characterized, it is not enough to refer that “the additional reagents were purchased from Sigma-Aldrich (Steinheim, Germany).”. For instance, acetylcholinesterases differ according to the source, and therefore inhibition also changes if you’re using the Eletrophorus electricus enzyme, the human enzyme, or extracted form an invertebrate. This information is lacking (here and in the section 2.5, where bioactivities should be properly described and referenced). The complete information (E.C. classification, origin of the enzyme, Sigma-Aldrich reference) is mandatory.

Answer: Reagents for enzyme inhibition activities have been presented in sections of ‘chemicals’ according to your suggestions.

Comment: Line 111: Write “The pellet was solubilized” instead of “The pellet was subtilized”

Answer: It has been changed.

Comment: Line 125: the authors mention antioxidant assays (ABTS, ORAC, FRAP), but not the methods and references; actually, no antioxidant results are presented either, so why mention this activity?

Answer: All information about antioxidant acitvity has been removed from the text. These results were included in draft, but due to the fact that classic antioxidant assays (ABTS, ORAC, FRAP) are now considered to have less predictive value when it comes to testing the role of dietary components in disease prevention in final version antioxidant potential has been removed.

Comment: Lines 121-127: The use of positive controls for the bioactivities is essential, since it allows the comparison of the results obtained for the samples with standard molecules. The authors do not mention these controls (and indeed there are no results for controls is Table 4). In addition, the concentrations used for the assays (in the order of mg/mL) are too high to be of interest.

The enzyme inhibitory activities (table 4) are too high to be of interest, the IC50 values presented in this work mean that all the samples should be considered as having no activity. In our lab we never go above 150 µg/mL, i.e., 0.150 mg/mL. Concentrations higher than that should not even be tested just to find an effect

Answer: Positive controls have been presented. Material and methods section has been supplemented with this information - The acarbose were applied as positive control in the case of α-amylase and α-glucosidase, in turn the Orlistat was used as a positive control for pancreatic lipase. They were used to monitor the correctness of the reaction, therefore the results are not included in Table 4 (< 5 mg/ml).

In the case of the anti-aging method, no reference sample was used due to the fact that Ferreres et al. (2010) did not indicate such a necessity. In these methods are sample control, where 50 mMol Tris-HCl pH 7,5 with 0,1 mol NaCl and 0,02 M MgCl*6H2O was used instead of the plant extract.

In the opinion of the authors, the presented results are interesting because they provide not yet published information on the bioactive potential of different varieties of carrots. The article does not state that it is a cure for diabetes or obesity, but shows some trends and differences between varietied. Of course, as already mentioned, carrots do not have such a strong health-promoting potential as isolated compounds or pharmaceutical preparations. However most of carrot extracts have higher IC50 for α-amylase, than sour cherry, red grapefruit, pineapple, orange or kiwi. Also IC50 for α-glucosidase are higher than for chokeberry, apple, pear or blackberry, a group of fruits recognized as important in the prevention of diabetes (Podsędek et al., 2014), which seems interesting though.

Comment: Line 216: the caption should be “3.3. Inhibitory activities of digestive enzymes by coloured carrot roots” instead of “3.3. Inhibitory activities of coloured carrot roots through digestive enzymes”

Line 219: acetylcholinesterase (AChE), and butyrylcholinesterase (BuChE) are not digestive enzymes, so they should not be included under “digestive enzymes” ; a suggestion is using a different caption in 3.3, such as “Enzyme inhibitory activities by coloured carrot roots”, since that would include all the enzymes studied.

Answer: It has been changed.

Comment: Lines 219-221: IC50, as correctly defined in Materials and Methods (line 125), is not “the lowest concentration of carrot that showed the highest inhibition potential”, but the concentration that inhibits activity by 50%. This must be corrected.

Answer: It has been changed.

Comment: Where are the results of antioxidant activities (ABTS, ORAC, FRAP) referred in Materials and Methods?

Answer: All information about antioxidant acitvity has been removed from the text. These results were included in draft, but due to the fact that classic antioxidant assays (ABTS, ORAC, FRAP) are now considered to have less predictive value when it comes to testing the role of dietary components in disease prevention in final version antioxidant potential has been removed.

Comment: I suggest the authors rewrite this work focusing only on the phytochemical and nutrient characterization of the carrot varieties, which is already interesting and useful, besides carrying out the necessary corrections.

Answer: The manuscript has been revised. All of them give a more complete screening about different varieties of carrots and allow to achieve the aim of the work.

Reviewer 2 Report

The comments are provided in the attached report.

Author Response

Thank You for Your review. Below are presented the answers to your suggestions

Comment: However, the introduction requires further clarification on carrots varieties/cultivars in different colours and whether the different sizes are related to duration of growth/harvesting time or its variety/cultivar. In addition, it is also required more extensive literature review regarding chemical compositions and nutrition aspects of different carrot varieties (colour and size), for instance, protein (amino acids), ash (vitamins and minerals), carbohydrate (fibre), fat.

Answer: The relevant information has been addend in Introduction part and also in Materials and Methods.

„..Cultivated carrots are grouped for root colour, sugar-carotenoid content, and root shape which they affect development period, temperature during this time and as well as fertilizers [4, 5]. Other reason for colour change in carrots is carotenoids that young carrots begin to accumulate after their first month of the growth and maintain about until the secondary growth concluded [6].  Carrot varieties range from 5 cm to 50 cm for their root lengths which are an important parameter for the marketing of carrot cultivars. Besides, cultivated carrots are classified as eastern and western carrots. Eastern carrots are purple and yellow coloured with branched roots; western carrots are orange, red and white with unbranched roots. In this case, eastern carrot cultivars are rich with anthocyanins and western carrots are abundant with carotenes [7]. Nevertheless, purple carrot is twice as rich with α- and β-carotene contents as orange carrot, moreover purple carrot possesses a sweet flavour with low total sugar content (REF-31). Thus, these stunning features of purple carrot make it a good alternative to orange carrot...”

„..Moreover, carrot is rich with trace mineral molybdenum which is crucial for metabolismes of carbohydrates, fats and iron absorption [20]. Besides, purple carrot contains 0 of fat, 31 g of carbohydrates with 1 g of fiber, and 1 g of protein; orange carrot includes 0.2 g of total fat, 6.9 g of carbohydrates with 2 g of fiber, and 0.7 g of protein…”

„…Carrot samples were grouped according to their sizes and colours. “Normal size” carrot in diameter (d) were between 20 mm and 45 mm and they weighed (m) from 50 g to 150 g; “mini size” carrots – 20 mm>d>10 mm and 50 g >m> 8 g;   “micro size”- d< 10 mm and m< 8 g….”

Comment: Please check the language used to be consistence, there are mixture between British and

American spelling throughout the article.

Answer: The language has been improved. The manuscript is after linguistic correction. If You would like to check it, please contact with office - translmed@translmed.com

Comment: Authors reported phenolic (phenolic acid, anthocyanin) and carotenoid contents are varied depending on the carrot size, what is the explanation of this? Please provide further  discussion regarding what factor influences the phenolic contents in different sizes of carrots

Answer: The extended discussion has been provided.

„…Thus, the differences in phenolic acid profiles of carrots might be related to the root structures size, colour and chemical compositions of the carrot varieties. Vice versa, orange, purple ,yellow and white carrots are results of phenolic and carotenoid contents during the maturation period as well as storage temperature [31, 32]…”

„…To summarise the polyphenolic contents of coloured carrot varieties, the highest total phenolic contents were found in NPC, MiPC, and MiOC. Thus, purple carrot has the highest total content of polyphenols, which was also confirmed by other authors [28, 35, 38-40]. Moreover, phenolic contents may change depending on genes, environment, and climate factors [41]…”

Comment: Line 172-173, authors stated that MiWC which had high in procyanidin contents and DP values and then jumps into conclusion that ‘the MiWC showed high health-promoting activities’. I don’t agree with that, it would require further support information from bioactivity assessment before making such statement. Therefore, this statement should be revised.

Answer: This part has been revised and the phrase " the MiWC showed high health-promoting activities" has been removed.

Comment: Surprisingly, the authors reported that no β-carotene detected in orange carrots (line 199). Please provide explanation for that.

Answer: The necassary discussion has been added. For us it was also surprising, therefore the analysis was repeated several times and the obtained chromatograms were compared with the standards. Finally, we are sure that the presented results are in line with the facts.

„… Carotenoids are instable pigments especially, during the work with them, for instance, orange carrot is known to be a rich source of β-carotene and α-carotene [43]. However, in the present study, α-carotene was not detected in MOC; moreover, according to the applied method, and retention time differences β-carotene was not found in NOC, MiOC, and MOC. It is surprisingly observation, but confirmed by the standard of β-carotene. Maybe the content was very low, therefore no peak of this compound was observed . Moreover, α- and β-carotene can be pressed by chlorophylls and xanthophylls as well [44]. Besides, storage period and temperature during the storage are other important factors for pigmentation…”

Comment: Extensive discussion requires on the enzyme inhibition potencies in the present study compared to other food samples/substrates, if there has not been previously reported in carrots. How these activities link to the phenolic and carotenoid contents of the individual sample? For instance, MiWC and MOC that showed the highest α-amylase activity. What are exactly mechanisms of action of phenolics and carotenoids on inhibiting the enzymes of interest?

- Is there any other bioactive components in carrots that might be associated with the bioactive properties reported herein?

Answer: The requested discussion sections have been provided.

„…Most of carrot extracts have higher IC50 for α-amylase, than sour cherry, red grapefruit, pineapple, orange or kiwi [26]. To clarify, α-amylase inhibitory activity is a result of bioactive compounds of plants such as glycosides, polysaccharides, steroids and terpenoids [50]. Overall, mini carrots showed the highest inhibitory effect against α-amylase…”

„…Therefore, phenolics are correlated with diabetes-related enzymes [54], however, in the present study was not determined high positive correlation in carrot varietes...”

„…The IC50 value for α-glucosidase are higher than for chokeberry, apple, pear or blackberry, a group of fruits recognized as important in the prevention of diabetes [26]…”

„…According to Fabroni [56], total anthocyanin content correlates with pancreatic lipase activity, however, in the present study, similar results were not observed. Since normal-sized orange carrot exhibited elevated activities against pancreatic lipaseMoreover, in the literature lentil cultivars have shown potent activity (IC50 from 6.26 to 9.26 mg/mL) against pancreatic lipase [57]. Hence, phenolics, ascorbic acid and carotenoids are responsable for pancreatic lipase inhibition [58]…”

Specific comments:

Comment: Line 32-33: Please clarify different colour/varieties carrots and define the different sizes

Answer: The information has been added in new version of manuscript (Materials and Methods).

Comment: Line 51: “…cancer, and aging…” and neuroprotection”, please remove ‘and’

Answer: It has been removed.

Comment: Line 55: suggest changing ‘includes’ to ‘contains’.

Answer: It has been changed.

Comment: Line 56: suggest changing ‘used’ to ‘able’

Answer: It has been changed.

Comment: Line 60: Please clarify ‘mini (baby) carrots’ are shorter growth or different variety

Answer: It has been explained.

„…Thus, to attract consumers, manufacturers prepare mixed carrot bags of yellow, purple, white, and orange carrots and call them “rainbow carrots”; moreover, mini (baby) carrots that range about 5 cm roots were created for consumption by young generations…”

Comment: Line 64: ‘in vitro’ must be in italic

Answer: It has been changed.

Comment: Line 90: suggest changing ‘removed’ to ‘mixed’; from ‘ml’ to ‘mL’

Answer: They have been changed in the whole manuscript.

Comment: Line 91: suggest changing ‘engineered’ to ‘performed’

Answer: It has been changed.

Comment: Line 92: please provide ‘sonication’ conditions

Answer: It has been added.

Comment: Line 93: ’19,000g’, the ‘g’ force must be in italic; change from ‘Hydrophillic’ to ‘hydrophilic’

Answer: They have been changed.

Comment: Line 94: suggest changing from ‘applied’ to ‘used’

Answer: It has been changed.

Comment: Line 94: ‘Triplicate trials were performed’, please confirm if the ’trials’ refer to extraction? If it is the case it would be more specify to use ‘The extraction was performed in triplicate.’

Answer: They have been changed.

Comment: Line 96: …tests were accomplished depending on…, please change to ... analyses were performed according to...

Answer: The sentence has been revised.

Comment: Line 98: please clarify ‘were finalized in 15 min’; change ‘flow rates’ to ‘flow rate’

Answer: They have been changed.

Comment: Line 98: please confirm if the condition was performed using gradient or isocratic

Answer: The injection and elution of the samples (5 μl) were concluded in 15 min with a sequence of linear gradients, but the flow rate was constant - 0.42 mL/min. This sentence was written more clearly

Comment: Line 102-103: merge two sentences to ‘…polymeric procyanidins was determined according to Kennedy & Jones [19].’

Answer: It has been changed.

Comment: Line 104: suggest changing ‘demonstrated’ to ‘expressed’

Answer: It has been changed.

Comment: Line 107: Please clarify ‘The powder of roots (0.20 g) containing 10% MgCO3 and 1% butylhydroxytoluene (BHT)…’

Answer: It has been changed. „To obtain the extracts for the determination of carotenoids, the lyophilized carrot powders ( ~ 0.20 g) containing 10% MgCO3 and 1% butylhydroxytoluene (BHT)…”

Comment: Line 110: suggest changing ’19,000g’ to ’19,000g’

Answer: It has been changed.

Comment: Line 111: please define ‘subtilized’

Answer: It has been changed to solubilized.

Comment: Line 113: suggest changing ‘realized’ to ‘analysed’

Answer: It has been changed.

Comment: Line 116: suggest changing ‘flow rates’ to ‘flow rate’

Answer: It has been changed.

Comment: Line 112; please clarify how the samples were prepared for each analysis

Answer: The preparation of samples has been presented.

“To obtain samples for the determination of carotenoids and chlorophylls, the lyophilized carrot powders ( ~ 0.20 g) containing 10% MgCO3 and 1% butylhydroxytoluene (BHT) were shaken with 5 mL of a ternary mixture of methanol/acetone/hexane (1:1:2, by vol.) at 300 rpm (DOS-10L Digital Orbital Shaker, Elmi Ltd., Riga, Latvia) for 30 min in the dark to prevent oxidation. The samples were centrifugated (4 °C, 7 min at 19,000g; MPW- 350, Warsaw, Poland), and recovered supernatants were acquired after the 4 times re-extracted from solid residue. Combined fractions were evaporated. The pellet was solubilized using methanol, and filtered through a hydrophilic polytetrafluoroethylene (PTFE) 0.20-μm membrane (Millex Samplicity® Filter, Merck, Darmstadt, Germany)…”

Comment: Line 117: please clarify the sentences ‘The operation was observed at 450 nm. The PDA spectra were  estimated over the wavelength range of 200–700 nm in steps of 2 nm.’ and specify the standards used for analysis (line 118)

Answer: The information has been reordered.

„…The elution solvents were linear gradient of acetonitrile:methanol (70:30%, v/v) (A) and 0.1% formic acid (B) as flow rates of 0.42 mL/min. The analysis was performed at 450 nm (carotenoids) and 660 nm (chlorophylls). The obtained spectra and retention times were compared with the authentic standards and in this way were determined the bioactive compounds. The tests were implemented in triplicate, and the results were presented as mg per kg of dm…”

Comment: Line 125: please spell out ‘ABTS, ORAC, FRAP’ for the first time of mentioning

Answer: This information has been removed. The results of the antioxidant activity were removed during the editing of the final version of manuscript. Information on this aspect in Materials and Methods is due to the inattention of the authors and has been removed.

Comment: Line 135: Sentence mentioned Table 2 which should appears in numerical order, therefore, suggesting to change Table numbers between Tables 1 and 2

Answer: It has been changed.

Comment: Line 141: Please specify the samples used in the literature

Answer: They have been specified.

„…The identified phenolic acids showed similarities for purple, yellow, orange and white carrot varieties to those reported in the literature [29]….”

Comment: Line 154: Please define the term ‘structures of the carrots’

Answer: It has been defined, please check in the revised version.

Comment: Line 157: Please clarify the term ‘similarities’ in amount or the compounds?

Answer: It has been changed. „…The identified anthocyanins showed similar compounds to those reported in the literature [34-37]…”

Comment: Line 178: Suggest to modify the table legend to ‘Overall phenolic compounds identified in 12 carrots varieties’; please clarify the numbers in column ‘MS/MS (m/z)’ as they are presented in different format

Answer: Title of table has been modified and different parts of MS/MS (m/z) have been removed from the manuscript.

Comment: Line 182: Table footnote, please change ‘…were not significantly different (p ≤ 0.05)…’ to ‘‘…were not significantly different (p > 0.05)…’

Answer: It has been modified.

Comment: Line 192-193: Suggest arranging the sample list which are detected the β-carotene and α-carotene by presenting from the highest to the lowest levels

Answer: It has been changed.

„…In the present study, β-carotene was detected in NYC > MYC > MiYC > MPC > NWC, while α-carotene was observed in NYC > MiOC > NOC > MiYC = NWC > MiWC…”

Comment: Line 210: suggest changing ‘…showed better results for total carotenoid contents.’ to ‘showed greater total carotenoid contents’

Answer: It has been changed.

Comment: Line 212: change from ‘MPCs’ to ‘MPC’

Answer: It has been changed.

Comment: Section 3.3 suggest changing ‘through’ to ‘against’

Answer: It has been totally changed. New version - Enzyme inhibitory activities by coloured carrot roots

Comment: Line 219: ‘in vitro’ should be in italic format

Answer: It has been changed.

Comment: Line 220: suggest changing ‘measured’ to ‘expressed’; remove ‘that’

Answer: They have been changed.

Comment: Line 240: Table footnote, please change ‘…were not significantly different (p ≤ 0.05)…’ to ‘‘…were not significantly different (p > 0.05)…’

Answer: It has been changed.

Comment: Line 245: please clarify the term ‘nutritional contents’

Answer: It has been changed as chemical content.

Comment: Line 254: please insert citation regarding the pancreatic lipase inhibitory activity of Indian spinach and suggest to include the quantitative value along with the statement

Answer: In the defined place other litarature information has been cited which is much more appropriate for that diccussion part. Check in the revised version, please.

Comment: Line 260: Table 4 legend, ‘in vitro’ must be in italic

Answer: It has been changed.

Comment: Line 264: Table footnote, please change ‘…were not significantly different (p ≤ 0.05)…’ to ‘‘…were not significantly different (p > 0.05)…’

Answer: It has been changed.

Comment: Line 294: please define the term ‘product quality’

Answer: It has been changed as nutrients.

Reviewer 3 Report

The manuscript describes the dereplication of carrot extracts, using UPLC-PDA and LC-MS-Q/TOF techniques that were established previously. The phytochemical profiles were correlated with antioxidant activity and a number of enzyme inhibition assays.

As such, the work is an extension and improvement of earlier work done on differently coloured varieties of carrot, e.g.:
Leja et al. (2013) The Content of Phenolic Compounds and Radical Scavenging Activity Varies with Carrot Origin and Root Color. Plant Foods Hum Nutr 68: 163–170. https://doi.org/10.1007/s11130-013-0351-3

The results are worth reporting, but at times the claims that are made are exaggerated and possible an opportunity was missed to get more detailed insight into the potential health properties of carrots and carrot varieties.

Claims: The polyphenols from carrot are described as vital for healthy human body functions such as protection against diabetes, cardiovascular diseases, osteoporosis, asthma, cancer, and aging and neuroprotection. Furthermore, polyacetylenes are used to destroy malignant cells such as leukaemia, myeloma, and lymphoma cells.

I can see where such a narrative comes from, but it is highly contested. Ok, polyphenols are widely considered to play a role in the prevention of degenerative diseases, but the mechanisms are far from being elucidated.

A direct reduction of reactive oxygen species (ROS) by scavenging activity of dietary antioxidants is now considered an obsolete idea; more likely is an effect on human cell signalling resulting in the activation of antioxidant pathways. This paradigm shift means that classic antioxidant assays (ABTS, ORAC, FRAP) are now considered to have less predictive value when it comes to testing the role of dietary components in disease prevention.

Considering the role of polyacetylenes: they may be toxic at certain concentrations, but nowhere are they used to destroy malignant cells. Falcarinol and falcarindiol may inhibit cell growth in experimental conditions, but no mechanism of action is known, no clinical uses of polyacetylenes are known.

Antidiabetic effects of carrots: it is true that patients with type-2 diabetes may be prescribed acarbose (an α-amylase inhibitor), but that doesn’t necessarily mean that all inhibitors of carbohydrate digestion are antidiabetic products. Sure, it may help reduce the intake of carbohydrates, but how many carrots would we have to eat to get an effect similar to that of prescription acarbose? It should be able to calculate that, and then if the amounts are not excessive, you might consider the antidiabetic effect of a carrot-based diet.

Anti-ageing effect of carrots: again, acetylcholinesterase is thought to play a role in onset of Alzheimer’s disease, and Alzheimer’s disease is linked to ageing. So, to call all acetylcholinesterase inhibitors anti-ageing agents is a bit over-the-top.

I would advise to tone down the exaggerated claims because they undermine the seriousness of the manuscript.

In my opinion, there was a missed chance to pinpoint which of the many compounds that are present in carrots may cause beneficial effects. As shown in Table 4, rather than testing individual compounds, crude extracts of carrots were tested in the enzyme assays. Therefore, in the end we do not know which compound we should focus on e.g. if we wish to inhibit lipase activity. Each carrot variety contained a mix of compounds, and the conclusions therefore by necessity remain speculative. Nevertheless, the results can be presented as a first crude screening exercise that may be followed up by more detailed experiments in the future.

Author Response

Thank you for your review. Below are presented the answers to your suggestions

Comment: As such, the work is an extension and improvement of earlier work done on differently coloured varieties of carrot, e.g.: Leja et al. (2013) The Content of Phenolic Compounds and Radical Scavenging Activity Varies with Carrot Origin and Root Color. Plant Foods Hum Nutr 68: 163–170. https://doi.org/10.1007/s11130-013-0351-3

Answer: For the literature search of coloured carrot varieties the study Leja et al. (2013) has been checked as well. The Leja’s study includes only phenolic compounds. In the present study, we investigated carotenoid contents, in-vitro enzyme activities as well as polyphenolic contents (some of them were identified for the first time) of carrot varieties. Apart from, in this work the color of the carrot and its various sizes were analyzed - normal, mini, micro. Therefore, the present work is definitely different from the others presented so far and provides a lot of new information both about bioactive compounds and the potential of health-promoting carrots.

Comment: Claims: The polyphenols from carrot are described as vital for healthy human body functions such as protection against diabetes, cardiovascular diseases, osteoporosis, asthma, cancer, and aging and neuroprotection. Furthermore, polyacetylenes are used to destroy malignant cells such as leukaemia, myeloma, and lymphoma cells. I can see where such a narrative comes from, but it is highly contested. Ok, polyphenols are widely considered to play a role in the prevention of degenerative diseases, but the mechanisms are far from being elucidated.

Answer: The introduction has been changed according to your suggestions with lower probability. Check the revised version, please.

“...Additionally, polyphenolic compounds are useful for healthy human body functions, supporting protection against diabetes, cardiovascular diseases, osteoporosis, asthma, cancer, aging and neuroprotection [16]..”

“...Moreover, carrot contains polyacetylenes, which might be able to destroy malignant cells such as leukaemia, myeloma, and lymphoma cells. Additionally, carrot contains luteolin that could protects against age-related symptoms in the brain, but the mechanisms are far from being elucidated [21, 22]...”

Comment: A direct reduction of reactive oxygen species (ROS) by scavenging activity of dietary antioxidants is now considered an obsolete idea; more likely is an effect on human cell signalling resulting in the activation of antioxidant pathways. This paradigm shift means that classic antioxidant assays (ABTS, ORAC, FRAP) are now considered to have less predictive value when it comes to testing the role of dietary components in disease prevention.

Answer: The results of the antioxidant activity were removed during the editing of the final version of manuscript. Information on this aspect in the abstract and purpose is due to the inattention of the authors and have been removed.

Comment: Considering the role of polyacetylenes: they may be toxic at certain concentrations, but nowhere are they used to destroy malignant cells. Falcarinol and falcarindiol may inhibit cell growth in experimental conditions, but no mechanism of action is known, no clinical uses of polyacetylenes are known.

Answer: We have just provided a general information about polyacetylenes not details about inhibition activities of falcarinol and falcarindiol. Besides, we agree with you that there is no clinical trial on polyacetylenes.

Comment: Antidiabetic effects of carrots: it is true that patients with type-2 diabetes may be prescribed acarbose (an α-amylase inhibitor), but that doesn’t necessarily mean that all inhibitors of carbohydrate digestion are antidiabetic products. Sure, it may help reduce the intake of carbohydrates, but how many carrots would we have to eat to get an effect similar to that of prescription acarbose? It should be able to calculate that, and then if the amounts are not excessive, you might consider the antidiabetic effect of a carrot-based diet.

Answer: We agree with your suggestion, but we are not able to make clinical studies for this aspect. Carrots are not a medicine and its consumption may only be prophylactic, not treating chronic noncommunicable diseases. Most of carrot extracts have higher IC50 for α-amylase, than sour cherry, red grapefruit, pineapple, orange or kiwi. Also IC50 for α-glucosidase are higher than for chokeberry, apple, pear or blackberry, a group of fruits recognized as important in the prevention of diabetes (Podsędek et al., 2014). Obviously, these results are much lower than in the case of isolated compounds or pharmaceutical preparations. However, they do provide some important screen regarding bioactivity properties of carrots.

Comment: Anti-ageing effect of carrots: again, acetylcholinesterase is thought to play a role in onset of Alzheimer’s disease, and Alzheimer’s disease is linked to ageing. So, to call all acetylcholinesterase inhibitors anti-ageing agents is a bit over-the-top.

Answer: Whole manuscript has been edited to not to be over-the-top. Please, check new version.

Comment: I would advise to tone down the exaggerated claims because they undermine the seriousness of the manuscript.

Answer: All claims have been revised regarding to your suggestions.

Comment: In my opinion, there was a missed chance to pinpoint which of the many compounds that are present in carrots may cause beneficial effects. As shown in Table 4, rather than testing individual compounds, crude extracts of carrots were tested in the enzyme assays. Therefore, in the end we do not know which compound we should focus on e.g. if we wish to inhibit lipase activity. Each carrot variety contained a mix of compounds, and the conclusions therefore by necessity remain speculative. Nevertheless, the results can be presented as a first crude screening exercise that may be followed up by more detailed experiments in the future.

Answer: We agree with your suggestion for future studies.

Round 2

Reviewer 1 Report

Although the manuscript is improved, I still maintain my opinion about the IC50 values being too high. The argumente that other works published results is not valid, it only means that the results from those references should not have been accepted as being "high" or "strong", as the authors write.

 The english still needs corrections, especially in parts of the text that have been added in the more recent version. try using "Grammarly", even the free version is quite helpful for correcting faulty english.

I add more detailed comments:

  • Line 66: write “metabolism” instead of “metabolismes”
  • Line 75: “range” implies two extremes, this sentence needs changing; maybe “that measure up to 5 cm”, or “that are approximately up to 5 cm in size”
  • Lines 80-81: although I still maintain that these bioactivities should not be included in the paper for the reasons I mentioned in the previous review report, it is incorrect to state that the in vitro activities are “against diabetes, obesity, and age-related disorders”, when in fact they are “against enzymes related with diabetes, obesity, and age-related disorders”
  • Lines 150-151: write “acarbose was used as positive control in the case of α-amylase and α-glucosidase, while Orlistat was used as a positive control for pancreatic lipase” instead of “The acarbose were applied as positive control in the case of α-amylase and α-glucosidase, in turn the Orlistat was used as a positive control for pancreatic lípase”
  • Lines 152-153: the expression “antiaging activity” does not refer to cholinesterase inhibitors, which can be used to decrease cognitive impairment in Alzheimer’s Disease but not to slow aging processes, so it should not be used here. Antiaging refers to agents that: (1) inhibit the overexpression and induction of ECM-degrading enzymes; (2) decrease the formation and action of ROS and UV-induced damage; (3) increased the expression levels of antioxidant enzymes (namely glutathione reductase, catalase, and manganese superoxide dismutase); (4) stimulate fibroblast and keracinocyte renovation; (5) reduce inflammatory processes that contribute to age-related pathologies; (5) increase the lifespan in models such as Caenorabditis elegans
  • Lines 152-153: no control was used for AChe and BuChe activity (e.g., donepezil)? The authors do not indicate the control.
  • Line 187: remove “vice versa”
  • Line 235: write “unstable” instead of “instable”
  • Lines 283-284 and following comments on the IC50 values: the fact that these IC50 values are lower than other reported does not mean that they are acceptable, even if the sources (apples etc) referred are cited as having antidiabetic effects; the IC50 values reported by the other authors should not have been accepted either, and the antidiabetic effect of apples and other foods does not mean that the effect is related with these enzymes and not to other factors acting together (e.g., pectins, antioxidant agents and so on).
  • Lines 304-306: it is abusive to conclude that “phenolics, ascorbic acid and carotenoids are responsable for pancreatic lipase inhibition”, even if there is a correlation. In fact, IC50 values of mg/mL can not be considered as “potent activity”, even if the authors of the reference classify them as such.
  • Table 4: remove "anti-diabetic by…" and "antiaging by", the activities determined were inhibition of  "∝-amylase", "lipase", and so on.
  • Table 4 and text: where are the results of the controls (acarbose and orlistat)? if they were used at all, where are the results?
  • Considering cholinesterase inhibition, any IC50 values above 150 µg/mL should not be considered as active; if the authors want to presente the results of these activities, they must not exagerate the interest of results which have no practical effect.

Author Response

Comments for Reviewer 1

Thank You for Your review. Below are presented the answers to your suggestions

Comment: Although the manuscript is improved, I still maintain my opinion about the IC50 values being too high. The argument that other works published results is not valid, it only means that the results from those references should not have been accepted as being "high" or "strong", as the authors write.

Answer: We agree that IC50 values are high, however, the results are related to sample differences. These are the results of carrot varieties. Hence, in the manuscript, there are statements like - "the highest activity of the enzyme...", or "the lowest...", because different varieties of carrots showed a different effect. However, we decided to take into account your comments, so in the abstract and the summary, the following conclusions were finally presented:

“...However, their pro-health effects (anti-diabetic, anti-obesity, anti-aging) should not be seen in the inhibition of amylase, glucosidase, lipase, and cholinesterase. Probably the mechanisms of their action are more complex, and the possible health-promoting effect results from the synergy of many compounds, including fiber, phytochemicals, vitamins, and minerals. Therefore, it would be worth continuing research on different varieties of carrots.”

Comment: The English still needs corrections, especially in parts of the text that have been added in the more recent version. try using "Grammarly", even the free version is quite helpful for correcting faulty English.

Answer: "Grammarly" has been applied for language check. In addition, the manuscript is after linguistic correction. If You would like to check it, please contact with office - translmed@translmed.com

Comment: Line 66: write “metabolism” instead of “metabolismes”

Answer: It has been changed.

Comment: Line 75: “range” implies two extremes, this sentence needs changing; maybe “that measure up to 5 cm”, or “that is approximately up to 5 cm in size”

Answer: That has been changed as “that is approximately up to 5 cm in size”.

Comment: Lines 80-81: although I still maintain that these bioactivities should not be included in the paper for the reasons I mentioned in the previous review report, it is incorrect to state that the in vitro activities are “against diabetes, obesity, and age-related disorders”, when in fact they are “against enzymes related with diabetes, obesity, and age-related disorders”

Answer: The information has been changed as “against enzymes related to diabetes, obesity, and age-related disorders”.

Comment: Lines 150-151: write “acarbose was used as a positive control in the case of α-amylase and α-glucosidase, while Orlistat was used as a positive control for pancreatic lipase” instead of “The acarbose were applied as positive control in the case of α-amylase and α-glucosidase, in turn, the Orlistat was used as a positive control for pancreatic lípase”

Answer: The sentence has been changed as “acarbose was used as a positive control in the case of α-amylase and α-glucosidase, while orlistat was used as a positive control for pancreatic lipase”.

Comment: Lines 152-153: the expression “antiaging activity” does not refer to cholinesterase inhibitors, which can be used to decrease cognitive impairment in Alzheimer’s Disease but not to slow ageing processes, so it should not be used here. Antiaging refers to agents that: (1) inhibit the overexpression and induction of ECM-degrading enzymes; (2) decrease the formation and action of ROS and UV-induced damage; (3) increased the expression levels of antioxidant enzymes (namely glutathione reductase, catalase, and manganese superoxide dismutase); (4) stimulate fibroblast and keracinocyte renovation; (5) reduce inflammatory processes that contribute to age-related pathologies; (5) increase the lifespan in models such as Caenorabditis elegans

Answer: The expression “antiaging activity” has been changed to “the activities of cholinesterase inhibitors”.

Comment: Lines 152-153: no control was used for AChe and BuChe activity (e.g., donepezil)? The authors do not indicate the control.

Answer: In the case of the anti-aging method, no reference sample was used due to the fact that Ferreres et al. (2010) did not indicate such a necessity. In these methods are sample control, where 50 mMol Tris-HCl pH 7,5 with 0,1 mol NaCl and 0,02 M MgCl*6H2O was used instead of the plant extract.

Comment: Line 187: remove “vice versa”

Answer: It has been removed.

Comment: Line 235: write “unstable” instead of “instable”

Answer: It has been changed.

Comment: Lines 283-284 and following comments on the IC50 values: the fact that these IC50 values are lower than other reported does not mean that they are acceptable, even if the sources (apples etc) referred are cited as having anti-diabetic effects; the IC50 values reported by the other authors should not have been accepted either, and the antidiabetic effect of apples and other foods does not mean that the effect is related with these enzymes and not to other factors acting together (e.g., pectins, antioxidant agents and so on).

Answer: We agree with your comments for plant chemicals which show function together, therefore, the enzyme inhibition is not the only result of the anti-diabetic effect either.

Comment: Lines 304-306: it is abusive to conclude that “phenolics, ascorbic acid and carotenoids are responsible for pancreatic lipase inhibition”, even if there is a correlation. In fact, IC50 values of mg/mL can not be considered as “potent activity”, even if the authors of the reference classify them as such.

Answer: Maybe you are right, however future studies are necessary to prove the statement.

Comment: Table 4: remove "anti-diabetic by…" and "antiaging by", the activities determined were inhibition of  "∝-amylase", "lipase", and so on.

Answer: They have been removed.

Comment: Table 4 and text: where are the results of the controls (acarbose and orlistat)? if they were used at all, where are the results?

Answer: The results of the controls have been added under the Table.

Comment: Considering cholinesterase inhibition, any IC50 values above 150 µg/mL should not be considered as active; if the authors want to present the results of these activities, they must not exaggerate the interest of results which have no practical effect.

Answer: We agree with your comments, however, the study aims to find the best carrot variety. Thus, we should present all data which we obtained.
